# Utilization of borax and its impact on the income and the livelihood of miners and other stakeholders: A case of Uganda

James Natweta Baguma[1,2]*, Victoria Mukasa[2], D. K. Sekimpi[1,2], Daniel Sekabojja[1,2], Victoria Nabankema[1,2], Kamese Geoffrey[2], Eva Magambo[2], John Ssempebwa[1], Margrethe Smidth[3]

**1** Department of Disease Control and Environmental Health, Makerere University School of Public Health, Kampala, Uganda, **2** Uganda National Association of Community and Occupational Health (UNACOH), Kampala, Uganda, **3** Diálogos, Copenhagen, Denmark

* bagumajamesnat@gmail.com

## Abstract

### Background

Mineral wealth serves as a crucial driver of economic growth and infrastructure development in many countries, including Uganda. Artisanal and Small-Scale Gold Mining (ASGM) plays a significant role in enhancing incomes and livelihoods. However, ASGM remains largely informal in Uganda, not reflected in the country's Gross Domestic Product (GDP). Many methods of gold processing are available around the globe. The borax method has become one of the favorable methods for extracting gold due to its lower operating cost, higher recovery, human and environmental consideration. This study aimed to assessing the impact of borax utilization in gold processing on the income and livelihoods of miners and stakeholders in Buhweju, Busia and Kassanda districts in Uganda where the Free Your Mine project is being implemented.

### Methods

A cross-sectional study using mixed methods was employed between October 2022 and January 2023 with data collected from 161 miners through semi-structured questionnaires, Key Informant Interviews (KIIs), Focus Group Discussions (FGDs), and observations. Analysis was conducted using Atlas ti version 7.0 and Stata Version 15.0.

### Results

Findings indicate low adoption of borax and limited training coverage on the use of borax. While 80.1% of respondents saved money attained from gold mining activities, only 21.5% had investments. FGDs revealed that miners often resort to mercury

**Data availability statement:** All relevant data are within the paper and its Supporting Information files.

**Funding:** This study was funded by Dialogos, a non-governmental organization (NGO) based in Denmark. through the Uganda National Association of Community and Occupational Health. (UNACOH). The support from Dialogos was instrumental in enabling the successful implementation of the project activities.

**Competing interests:** The authors have declared that no competing interests exist.

use for quick income as using mercury takes little time compared to mercury-free methods, while KIIs emphasized the need for government intervention and need for policies to promote safer gold processing methods for sustainable livelihoods.

## Conclusion

Study underscores need for awareness, policy to improve safety in Uganda's ASGM sector. In addition, there is need for funding to support scale up the Free Your Mine Project to other gold mining districts in Uganda.

---

## Introduction

Globally, an estimated 100 million people depend on income from small-scale mining [1].In addition to providing a livelihood for thousands of households, small-scale mining reduces migration from rural to urban areas [2]. Small-scale gold mining provides jobs in remote villages, reduces the migration of able-bodied people to urban areas, and helps fight poverty.

In the Minamata Convention on Mercury, ASGM is defined as –gold mining conducted by individual miners or small enterprises with limited capital investment and production [3]. It is one of the emerging economic activities providing alternative livelihoods globally with more than 13 million artisanal and small-scale miners and about 150 million people indirectly dependent on it [4].

For more than fifty years Artisanal Small-scale Gold Miners (ASGMs) from Benguet in the northern part of the Philippines have used a mercury-free gold extraction method. They have realized that using borax instead of mercury yields more gold than any amalgamation method using mercury. Borax, also known as sodium borate is a mixture of detergents with many household cleaning functions. It is classified as non-toxic and causes no known chronic health effects [5]. It is mixed with a concentrate of the heavy metals collected through washing the ore, and then it is heated with a burner thus extracting the gold [6]. Some ASGMs together with BanToxics, a Nongovernment Organization (NGO) from the Philippines, and Dialogos, an NGO from Denmark implemented the mercury-free method in different mining areas of the Philippines [7,8] Diálogos has learned from previous projects that the road to better working and living conditions goes through forming organizations [8]. Formalization of the sector is another important step towards regulation of the working conditions. Likewise, the only way to sustainably secure mercury-free gold extraction is through legalization [9].

In the Free Your Mine 2019–2021 project ASGMs were introduced to the economic and health advantages of the mercury-free gold extraction method as an incentive to change to the mercury-free method. The strengths of the mercury-free gold extraction method include low costs, higher gold yield, benign environmental impact, and legality. The truth is the mercury-free method is less time-consuming than the mercury amalgamation method because the former uses wet milling meaning that the ore is fed into the ball mill soon after sorting the ore. On the other hand,

the mercury amalgamation method that starts by drying the ore in the sun may take a few hours to days depending on the weather which could be dry, cloudy, or wet. By the time the ore dries, the miner using the Gravity borax method has already extracted her/his gold. Borax is typically accessible in developed urban areas, as it is commonly used in the welding industry and by jewelers, however, it is hard to be found in remote villages. As access to borax is essential for the implementation of the mercury-free gold extraction method, the establishment of borax distribution in remote villages is beneficial [10].

This project also intended to reduce mercury pollution. Mercury's damages to human health includes pneumonia, failure of liver and kidney, memory loss, tremors, lethargy, insomnia and changes in an individual's personality [11]. Moreover, the effect on the environment includes the reduction of micro-biological activity vital to the terrestrial food chain in soils [12]. On the other hand, the use of borax causes insignificant effect to human health and environment compared to the use of mercury. The process shows to possess lower operating cost and higher recovery of gold as a result of no formation of mercury flour which leads to gold losses in a process [13]. It is against this background that a mercury-free gold extraction demonstration site was put up in each of these study areas in Buhweju and Kassanda to be centers for capacity building in mercury-free gold extraction.

One of the outcomes of the Free Your Mine Project was the formation of associations and groups by the ASGMs in the study districts. The formation of the associations and groups was intended to organize the miners to mine and carry out their activities in a responsible and organized manner [14].

Harmonization and reorganization of the mining activities in the project districts was top of the agenda as far as this project intervention was concerned. Feedback from the miners and local leaders has shown that in areas where associations and groups have been formed, there is harmony in the way the miners operate, lawlessness and other bad practices are on the decline and miners are looking at alternative methods to mercury use as well as ways to diversify their sources of income. The several capacity building meetings, trainings and engagements with the miners helped the miners to gain confidence to start forming groups, think of alternative ways to mercury use in gold mining and ways to survive on other sources of income [15].

Miners, community leaders, teachers, and healthcare workers were introduced to the mercury-free gold method [8]. At one event the local gold extraction method using mercury and the mercury-free gold method were compared in terms of gold recovery. The mercury-free method recovered 40% more gold than the mercury amalgamation method [8]. All agreed that the mercury-free method is a very good alternative to the mercury-dependent method [16].

Mining has played a substantial role in the development of many countries for example Ghana, which is second only to South Africa in terms of gold production on the African continent. Ghana is gifted with rich mineral resources and was formerly called the Gold Coast. Mining accounts for about 9.1% of Ghana 's gross domestic product (GDP) and employs almost 300,000 people [17,18]. Notwithstanding the role played by mining in the socioeconomic sector, mining has played an important role in the development of Ghana. Unfortunately, in Uganda, ASGM is still mostly in the informal sector and is not accounted for in the country 's GDP calculation. However, it is estimated that if it is included in the formal sector, Uganda 's GDP would increase by more than 2%. Equally in Uganda, ASGM employs approximately 300,000 people directly and approximately a million people benefit from it indirectly [18,19]. Like all industries, mining has both benefits and risks for the people living in communities where minerals are found. Management of the environmental and health impacts can improve the lives of the people in the community. However, it is estimated that if included in the formal sector, Uganda's GDP would increase by more than 2% [20]. Uganda majorly depends on agriculture as the driver of the economy. In Buhweju, Busia and Kassanda, majority of the population engage in crop and animal farming on a small scale as a source of income and livelihood [21]. Practicing mercury-free gold processing with proper consideration of environmental and human health can have positive or negative impacts on the income and livelihoods of the Artisanal miners as well as the stakeholders involved in the gold mining value chain.

To date, only a few reports have been published on the impact of borax use in gold processing on the economy and livelihood of Ugandans; the majority of the studies are mainly tailored to the health effects related to artisanal and small-scale gold mining and do not offer detailed information on the income and the livelihood of the miners and other stake-holders in Uganda. Thus, information regarding the adverse effects of ASGM mining on the income and livelihoods of communities within mining areas is insufficient. This study was conducted to ascertain the impact of borax use on the income and livelihoods of the gold miners and the communities around them (stakeholders) in selected gold mining districts Uganda. Findings from this study can help to establish the parameters of future income and livelihood-related surveys in ASGM communities.

## Materials and methods

### Study place, population, and design

The study was conducted in three gold mining districts in Uganda – Buhweju, Busia, and Kassanda – which had active ASGMs as shown in Fig 1 below. These districts are located in three different regions where gold deposits are found namely Buhweju in the Western region, Busia in the Eastern region where some mining sites are located close to the shores of Lake Victoria, and Kassanda in the Central region. The study population included stakeholders at the national level in the gold mining sector, District leaders, leaders of associations of ASGMs and management committees of demon-stration/ training sites, gold buyers/ mercury sellers, borax dealers, and ASGMs. A cross-sectional study employing both qualitative and quantitative methods was conducted between October 2022 and January 2023. The study included key informant interviews (KIIs), focus group discussions (FGD), as well as observations at mining sites, business assessment checklists, and administration of face-to-face semi-structured and structured questionnaires.

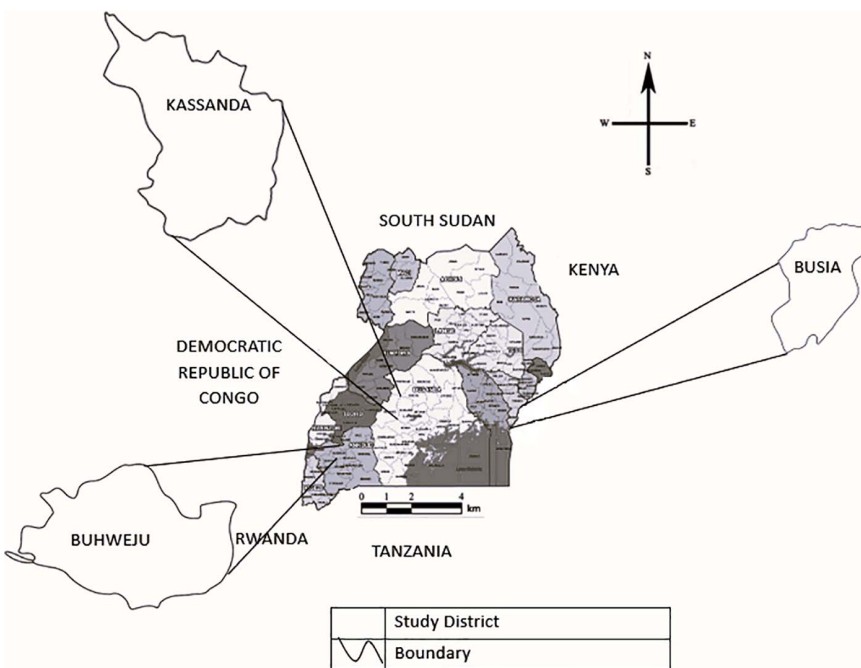

**Fig 1. Study districts.** PlaniGlobe, http://www.planiglobe.com, CC BY 2.0.

## Quantitative sample size determination

The sample size was determined using Lesley Kish's [22] formula considering a conservative prevalence of 12% prevalence of uptake and compliance to mercury-free gold extraction and health, safety and environment practices among ASGMs which was carried out in Burkina Faso [23]. A sampling error of 5% was considered giving a sample size of 161 participants as shown below.

$$n = \frac{Z^2 P(1-P)}{d^2}$$

Where:
n - Estimated sample size
Z – Standard normal value corresponding to the level of confidence 95%(1.96)
P – Is the conservative prevalence = 12%
d – standard error (5%)

$$n = \frac{1.96^2 0.12(1-0.12)}{0.05^2}$$

$$n = 161$$

Stratified random sampling was employed in this study. Each gold extraction site was considered a stratum and the number of respondents in each stratum was determined by the formula:

$$\frac{\text{number of ASGMs at the gold extraction site}}{\text{total number of ASGMs in selected gold extraction sites}} \text{ x sample size}$$

Participants were randomly sampled from the list of ASGMs obtained from the mine site managers at the gold extraction site. A total of 53, 54 and 54 study participants were interviewed from Buhweju, Busia and Kassanda districts respectively.

## Qualitative sample size determination

Qualitative methods: An estimated 12 Key Informant Interviews (KIIs) were conducted in each mining district to establish the impact of borax use on the income and livelihoods of the miners. Other stakeholders and policymakers at the national level were contacted as stakeholders. The actual number of KIIs was determined by data saturation.

Three FGDs with 8–10 participants who were directly involved in the gold mining process were conducted in each of the study districts one among the men, one among the women, and one among the youths. as shown in Fig 2.

## Sampling method and procedures

The quantitative sample estimates were obtained using probabilistic sampling techniques. The choice of the random stratified sampling approach to be adopted for the quantitative sample was guided by the data from recent studies and the current situation [23]. The qualitative respondents were selected purposively based on their knowledge and positions in the study district and the gold mining associations. Three mining sites were selected in each mining district including the Free Your Mine Project demonstration mining site, and two other sites were selected randomly from the list of mining sites provided by the district authorities.

## Dependent variables

Effects of mercury-free gold mining on the income and livelihoods of miners and other stakeholders.

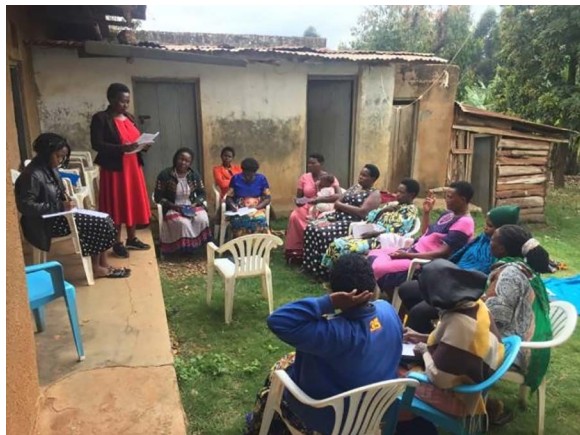

**Fig 2. Sharing experiences and insights in a focus group discussion on gold mining.**

### Independent variables

Location of the gold mining site, political environment, individual factors, socio-demographic factors, health safety factors, environmental factors, and business factors.

### Inclusion criteria

All ASGMs who had operated in the selected mine for at least 1 year and consented to participate in the study. In addition, we included district and national level stakeholders who are key in the ASGM sector.

### Exclusion criteria

Individuals in the mining districts who were not technically or administratively involved in the ASGM process either at the district or national level.

### Data Collection methods

The study team employed both quantitative, qualitative data collection tools in the conduct of this study.

### Quantitative data collection (survey)

Using a semi-structured questionnaire, 161 interviews were conducted using questionnaires pre-loaded on tablets. Kobo Collect software program was used to aid electronic data capture. The questionnaire had incorporated automated skip commands and constraints to keep logical flow of questions in the survey tool. The choice of Kobo Collect programming for this assignment was aimed at minimizing human error and non-entry of responses that are associated with manual entry. Further, the technique reduced time wastage on data entry with all the associated data entry clerk errors.

### Qualitative data collection and techniques

Key informant interviews were conducted using a key informant guide, for district officials (District Production Officer, District Health Officer, District Labor Officer, District Community Development Officer, District Health Inspector, the Chief Administrative Officer (CAO), the Local Council V (LC5) Chairperson, Gold buyer's/mercury sellers), borax dealers among others. National level stakeholders from the Ministry of Health, Ministry of Gender Labour and Social Development, NEMA, Directorate of Geological Survey and Mines, and Ministry of Education and Sports were also contacted as shown

in Fig 3. The Focus group discussion guide was used to aid facilitation of the FGDs. A previous community survey on a related topic recommended that facilitated focus group discussions may provide more a larger quantity and dependable information, as interacting with peers they trust is more likely to facilitate people's willingness and readiness to contribute to the subject matter [24].

## Quality control measures

Before the data collection, RAs were trained rigorously for two days. The study tools were then pretested using respondents from the neighboring mining sites not included in the study. Letters of introduction were provided to the research team to introduce them to the leadership in the participating districts and the mining sites. Appointments were also scheduled with the district leaders and leaders of mining sites where possible.

Consultation with business experts was done to ensure the validity of the study. During data collection, data collected was reviewed each day to check for errors in filling, completeness, legibility, and other consistency issues. Data was entered immediately after it was collected. The study tools were translated into the three local languages used in the selected mining areas. In Buhweju is Runyankore, in Busia it is Samya and in Kassanda it is Luganda. This ensured that the RAs asked the questions and the participants responded in their local language.

## Data management

The quantitative data was imported from the Kobo Collect tool and exported to Microsoft Excel for cleaning. Subsequently, the cleaned data was exported to Stata version 15 for thorough analysis and interpretation.

## Qualitative data analysis

For the qualitative data, the KIIs and FGD were recorded using audio recorders. These audio files were then transcribed verbatim by trained qualitative data assistants. Data was analyzed using the thematic content analysis method following the semantic approach where a codebook was developed after reading through the transcripts several times and coding conducted. Thereafter, similar codes were grouped to form sub-themes that formed the study themes. Data analysis was supported by Atlas ti version 7.0 software. Quotations were used to supplement the presentation of the study themes.

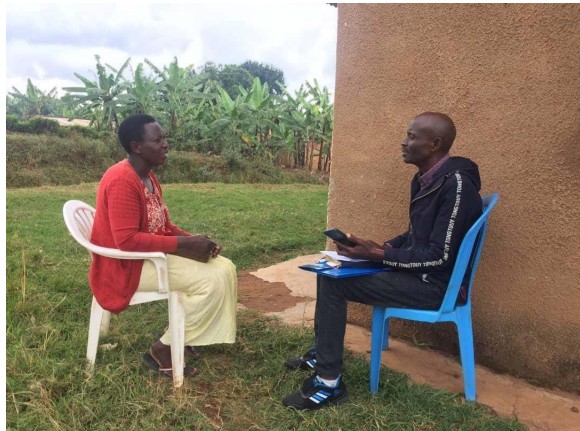

**Fig 3. Key Informant interview with the leader of the association of ASGMs and management committees in Gold mining in Buhweju.**

### Quantitative data analysis

The data were cleaned using MS Excel 2016 and analyzed using STATA 15.0 statistical software. We performed binary logistic regression to obtain crude odds ratios and their corresponding 95% confidence intervals (CIs). Variables with $p < 0.02$ were selected for inclusion in multivariate logistic regression to identify significant predictors for each outcome. Statistical significance levels were two-sided at $p < 0.05$.

### Ethics statements

Participation in the study among the artisanal small scale gold miners was voluntary. Research ethical approval to conduct the study was obtained from the Makerere University School of Health Sciences Institutional Review Board (MakSHS-IRB No-MAKSHSREC-2022–346. The study was submitted to the Uganda National Council for Science and Technology (UNCST) for approval No-HS2405ES. After approval by UNCST, study objectives were clearly explained to the participants before obtaining written informed consent from all participants. The participants who could not write had their fingerprints recorded/taken. In addition, the individuals in this manuscript have given written informed consent. The names of the participants were not recorded. The data collected was stored on drives that were only accessed by the principal investigator and the data manager.

## Results

### Socio-demographic characteristics

Of the 161 participants, more than three quarters, 77.6% (125/161) were males. Approximately two thirds of the respondents, 66.4% (107/161) were aged 18–35 years and only 0.6% (1/161) were aged above 65 years. The study revealed that the majority of the respondents had not completed the primary level 36.6% (59/161) followed by 25.5% (41/161) who had not completed the secondary level. More than half of the respondents were doing excavation or digging 52.8% (85/161) followed by 41.6% (67/161) who were panning or washing. It was noted that there was no borax seller 0.0% (0/161). The study also revealed that more than three-quarters of the respondents had worked in the gold mining and processing sector for 0–10 years 83.0% (133/161) as shown in Table 1 below.

### Attitudes

The study revealed that when the respondents were asked about what they prioritize when choosing a gold extraction method, it was noted that the majority of the respondents prioritized their health 52.8% (85/161) followed by the health of their family and co-workers, followed by the income, followed by work efficiency/ how fast and easy work is, followed by importance of environmental protection and lastly the cost of extraction method. It is important to note that environmental protection was the least prioritized as shown in Table 2 below.

**Borax use.** The study findings regarding borax use revealed notable gaps in awareness, training, and utilization among the respondents especially from the miners who were interviewed from the non-demonstration mining sites in the study districts. Specifically, it was found that a substantial portion, accounting for 57.1% (92 out of 161 respondents), had not been acquainted with the gravity borax method. Furthermore, 21.7% (35 out of 161 respondents), had not received any training in this method. Interestingly, only a minority, representing 16% (26 out of 161 respondents), reported using borax, with 13% predominantly utilizing it for gold purification and a mere 1.9% for gold extraction. Among those using borax, the primary reason cited was its perceived safety for both human health and the environment. In terms of quantity, the majority of users indicated utilizing 0–10 grams of borax. A considerable proportion of respondents in the non-demonstration sites demonstrated a lack of understanding regarding the advantages of borax in gold extraction, as evidenced by the data presented in Table 3.

**Table 1. Sociodemographic of the respondents.**

| Variable | Attribute | Frequency(n) | Percentage (%) |
|---|---|---|---|
| **Age in Complete years** | 18–35 | 107 | 66.4 |
| | 36–50 | 43 | 26.7 |
| | 51–65 | 10 | 6.2 |
| | >65 | 1 | 0.6 |
| **Sex** | Female | 36 | 22.4 |
| | Male | 125 | 77.6 |
| **Level of Education** | No formal education | 8 | 5.0 |
| | Primary level completed | 30 | 18.6 |
| | Primary level not completed | 59 | 36.6 |
| | Secondary level/high school completed | 14 | 8.7 |
| | Secondary level/high school not completed | 41 | 25.5 |
| | Tertiary Level (Diploma/Degree) | 9 | 5.6 |
| **Do you take Alcohol?** | No | 113 | 70.2 |
| | Yes | 48 | 29.8 |
| **Type of gold extraction work** | Excavator/digger | 85 | 52.8 |
| | Panner/Washer | 67 | 41.6 |
| | Carrier | 31 | 19.3 |
| | Crusher | 26 | 16.1 |
| | Burner | 17 | 10.6 |
| | Gold buyer | 15 | 9.3 |
| | Mercury seller | 2 | 1.2 |
| | borax seller | 0 | 0 |
| **How many years have you worked with gold mining?** | 0–10 | 133 | 83.0 |
| | 11–30. | 24 | 14.9 |
| | 31–50 | 4 | 2.5 |

**Table 2. Attitudes.**

| Variable | Attribute | Frequency(n) | Percentage (%) |
|---|---|---|---|
| **If you could choose a new gold processing method, how important is your income to you?** | 1st priority | 52 | 32.3 |
| | 2nd priority | 53 | 32.9 |
| | 3rd priority | 22 | 13.7 |
| | 4th priority | 6 | 3.7 |
| | 5th priority | 5 | 3.1 |
| | 6th priority | 5 | 3.1 |
| **If you could choose a new gold processing method, how important is the cost of extraction method to you?** | 1st priority | **32** | **19.9** |
| | 2nd priority | 38 | 23.6 |
| | 3rd priority | 34 | 21.1 |
| | 4th priority | 18 | 11.2 |
| | 5th priority | 9 | 5.6 |
| | 6th priority | 12 | 7.5 |

**Table 3. Borax use among the Artisanal small scale gold miners (follow-up questions used).**

| Item | Response | Frequency | Percentage |
|---|---|---|---|
| Have you heard about the gravity borax method for gold extraction? | No | 92 | 57.1 |
| | Yes | 69 | 42.9 |
| Have you been trained on the gravity borax method for gold extraction? | No | 35 | 21.7 |
| | Yes | 34 | 21.1 |
| Have you used borax to extract gold before? | No | 43 | 26.7 |
| | Yes | 26 | 16.1 |
| What did you use the borax for? | Clean dirty gold | 16 | 9.9 |
| | Purifying/ refining the gold | 21 | 13.0 |
| | Extract gold from the ore | 3 | 1.9 |
| What benefits or advantages have you gained (experienced) from using borax? | Cheap | 6 | 3.7 |
| | Clean weighs more | 4 | 2.4 |
| | Is not dangerous to humanand environmental health | 7 | 4.3 |
| How often do you use borax? | Daily | 1 | 0.6 |
| | Monthly | 9 | 5.6 |
| | Weekly | 8 | 5.0 |
| How much borax do you approximately use in a month? In grams | 0–10 | 11 | 6.8 |
| | 11–20. | 5 | 3.1 |
| | 21–30 | 1 | 0.6 |
| Mention any advantages of Use of borax in gold extraction that you know of. | Capture/get more gold | 11 | 6.8 |
| | Cheap | 20 | 1.4 |
| | it is mercury free | 23 | 14.3 |
| | Environmentally safer | 19 | 11.8 |
| | Easy to learn | 10 | 6.2 |
| | Purifies gold | 5 | 3.1 |
| | Don't know | 26 | 16.1 |

The study revealed that borax use is highly influenced by the inflow of mercury into the country by gold buyers who sell mercury to the miners who sell gold to them as indicated below;

*"I think the main solution is to ban where the mercury is coming from and that is Tanzania. When they ban where the mercury is coming from, then we are not going to get it and then also, they should give people access to borax. So that when people know that borax is close by, they can get it and use it. Another thing is that we are the same people who bring in mercury, we the gold buyers. When they sell the gold to you, they ask you for more mercury and you go to the market, get mercury, and give it to the miners but when it has been banned and you cannot get it, you have no alternative but to use borax to replace the mercury. We usually buy the mercury from Kampala and the Kampala people also get it from Tanzania." - Gold buyer Buhweju*

**Likelihood of continuing to use mercury after training on borax use.** Another logistic regression analysis examined the association between borax use (as a marker for mercury-free gold processing) and miner training in non-mercury methods. The results revealed that miners who had received training in the borax method were 5.43 times more likely to utilize borax compared to those who had not been trained (OR=5.43; 95% CI: 1.56–18.85). However, no significant association was found between the use of borax and the years of experience of miners in gold mining (OR=0.96; 95% CI: 0.90–1.03; p-value >0.05), as indicated in Table 4.

It is important to note that despite the various training in mercury free gold extraction using borax along with the health and livelihood improvement outcomes that come along with it, the artisanals have still not taken it up as indicated below.

*"People have not stopped using it, they are still using it because there is no alternative. People have been trained on use of borax but still they are not complying. There is need for government to come in and enforce because it has become a safety issue."KII Uganda Association of Artisanal and Small-Scale Miners (UGAASM)-Buhweju*

In addition, it was noted that there is need to increase coverage of training and incentivization among gold miners to increase uptake of the borax method as indicated in one of the key informants.

*"We need to train more miners in Uganda about the Borax method to interest all miners about the whole process. This is because miners are concerned about earning daily income and survival. More miners should be informed about wet milling method. They need to know more about the whole process"* **-KII Busia**

### Income and livelihoods of the Artisanal Small scale gold miners

It was revealed by the study that the majority of the respondents 68.9%(111/161) earn 100,001–500,000 Uganda shillings monthly from ASGM activities. 80.1% of the respondents were saving some of this money. The majority of them were using the saved money from gold mining activities was invested in avenues such as land and businesses 50.8%(82/181) and those who were not saving claimed that their expenses outweighed their incomes 43.5%(14/32) as shown in table 5 below.

Insights from stakeholders underscore the complex dynamics surrounding mercury use in Uganda's ASGM sector, shedding light on attitudes, economic considerations, and potential pathways towards safer mining practices;

### Facilitators to uptake of borax

Identifying facilitators to the adoption of mercury-free gold mining techniques involved understanding incentives, accessibility, and awareness. These factors play crucial roles in encouraging miners to embrace safer and more sustainable methods as shown in Table 6 below.

**Difference in the mean income between mercury users and borax users.** The average income earned by miners was Ugx. Sh. 436,770.4 (~USD 115) (SD±839622.3), Ranging from Ugx sh.1000 (~0.27 USD) to 10m (~2746USD)

**Table 4. Likelihood of continuing to use mercury after Learning about gravity/borax method, n = 161.**

| Ever heard about gravity concentration method | Unadjusted OR | P-value | [95% CI] |
|---|---|---|---|
| Currently use mercury gold extraction | | | |
| No | 1.0 | | |
| Yes | 4.83 | 0.000*** | (2.25-10.37) |
| Ever used mercury in gold extraction | | | |
| No | 1.0 | | |
| Yes | 1.13 | 0.001** | (1.05-1.23) |
| Currently use borax method in gold extraction | | | |
| No | 1.0 | | |
| Yes | 5.43 | 0.008** | (1.56-18.85) |
| Years of experience in gold mining | 0.96 | 0.283 | (0.90-1.03) |

**Table 5. Income and livelihoods of the Artisanal and small-scale gold miners.**

| Item | Response | Frequency | Percentage |
|---|---|---|---|
| How much do you earn on a monthly basis from ASGM activities? | 0–100,000 | 23 | 14.3 |
| | 100,001–500,000 | 111 | 68.9 |
| | 500,001–1,000,000 | 21 | 13.0 |
| | >1,000,000 | 6 | 3.7 |
| Do you save any of this money? | No | 32 | 19.9 |
| | Yes | 129 | 80.1 |
| If Yes, How much of this money do you save? | 0–100,000 | 73 | 45.3 |
| | 100,001–500,000 | 52 | 32.3 |
| | 500,001–1,000,000 | 1 | 0.6 |
| | >1,000,000 | 4 | 2.5 |
| How do you spend the money that you have saved from gold mining activities? | Investment (Land, Business) | 82 | 50.8 |
| | School fees | 53 | 32.9 |
| | Health costs | 47 | 29.2 |
| | Food | 50 | 31.1 |
| | Others(Farming and family) | 35 | 21.7 |
| If no to saving, why don't you save? | My expenses outweigh my income | 14 | 43.5 |
| | Don't know how/lack financial knowledge | 5 | 15.6 |
| | Have many debts | 2 | 6.3 |
| | Don't have a specific budget for expenses | 3 | 9.38 |
| | No personal goals | 0 | 0.0 |
| | Don't want/not interested | 8 | 25 |

**Table 6. Facilitators to the uptake of mercury-free gold mining methods.**

| Kassanda | Buhweju | Busia |
|---|---|---|
| -A crude saving system among miners' groups exists.<br>Effective Leadership Structures<br>Legal operation status for the miners | Presence of a mercury free gold mining demonstration sites<br>Capacity building on the use of borax and other economic empowerment skills<br>A legal saving system<br>(SACCO) with<br>appropriate<br>documentation among miners exists. And women miners have been trained in financial literacy. | Support from the District Local Governemnt<br>NGOs have trained miners in the mercury-free gold mining methods and financial management & savings. |

A T-test for the difference in the average income earned was also done, and although mercury users' mean monthly income was higher than for mercury-free users (a mean difference in a monthly income of Ugx. Sh.51,706)(~14.2 USD), this difference was not statistically significant, t-statistic=−0.370, pvalue<0.05. Results are displayed in Table 7 below.

**Two-sample t test with equal variances.**

**Mercury use.** About mercury use, It was noted that approximately two thirds of the respondents 65.8% (106/161) affirmed that they had ever used mercury for gold extraction and majority of them 62.1% (100/161) had used it in the last year (0–12) months. Almost half of them 43.0%(69/161) had used 0–10 grams. It was noted that majority of the mercury users buy it for themselves 45.3%(73/161) and most of them use it on a weekly basis. Majority of the respondents who were not using mercury 24.8%(40/161) mentioned that they were using water washing with no chemicals. Majority of the respondents said that they usually buy a gram of mercury at 1000 Uganda shillings as shown in Table 8 below.

**Table 7. *t*-test for difference in mean monthly income between mercury users and non-mercury users.**

|  | Group | Obs | Mean | Std. Err. | Std. Dev. | [95% Conf. Interval] |
|---|---|---|---|---|---|---|
| No | 55 | 402727.9 | 59195.11 | 439002.7 | 284048.9 | 521407 |
| Yes | 106 | 454434 | 95867.26 | 987013.9 | 264346.9 | 644521 |
| combined | 161 | 436770.4 | 66171.51 | 839622.3 | 306088.2 | 567452.6 |
| diff |  | −51706.05 | 139906.4 |  | −328020.7 | 224608.6 |

diff = mean(No) - mean(Yes), t = -0.3696, Ho: diff = 0.

degrees of freedom = 159.

Ha: diff < 0, Ha: diff!= 0, Ha: diff > 0.

Pr(T < t) = 0.3561, Pr(|T|> |t|) = 0.7122, Pr(T > t) = 0.6439.

**Table 8. Mercury use in Artisanal small scale gold mining.**

| Item on practice | Response | Frequency | Percentage |
|---|---|---|---|
| Have you ever used mercury for gold? | **No** | **55** | **34.2** |
|  | Yes | 106 | 65.8 |
| When is the last time you used mercury?(in Months) | 0–12 | 100 | 62.1 |
|  | 13–48 | 5 | 3.1 |
|  | >48 | 1 | 0.6 |
| How much mercury in grams did you use? | 0–10 | 69 | 43.0 |
|  | 11–100 | 26 | 16.0 |
|  | 101–1000 | 7 | 4.0 |
|  | >1000 | 4 | 2.5 |
| Where do you mostly get the mercury that you use? | Both buy and given for free | 8 | 5.0 |
|  | Buy for self | 73 | 45.3 |
|  | I get for free center | 24 | 14.9 |
| How often do you use mercury? | Daily | 31 | 19.3 |
|  | Monthly | 17 | 10.6 |
|  | Other (Please specify) | 10 | 6.2 |
|  | Weekly | 47 | 29.2 |
|  | Total | 161 | 100.0 |
| How much is a gram of mercury? (in shs) | <1000 | 17 | 10.6 |
|  | 1000 | 67 | 41.6 |
|  | >1000 | 15 | 9.3 |
| If you haven't used mercury before, what method are you using? | Borax | 2 | 1.2 |
|  | Water and other methods | 40 | 24.8 |

The qualitative data from Uganda's ASGM sector reveals challenges and attitudes towards mercury use. Stakeholders express concerns about health effects but cite economic necessity. Proposed solutions include banning mercury importation, promoting alternative methods like borax, and integrating new methods alongside mercury for those who prefer it. As shown below;

*"We have tried to sensitize the public/communities but the people have a negative attitude, they prefer health effects to poverty. So, the government should hurry to bring alternatives."L.C1 Kagaba*

*"Mercury is affordable for even poor to get gold. Other methods like wet pan are for rich, its expensive, yet mercury like 10 gram I can get at UGX 7,000/= (~1.96 USD)and use to get like UGX 100,000/=(~27.46 USD) from gold."-ball mill manager, Kagaba, Kassanda district*

The study revealed that the artisanal small scale gold miners are still finding it hard to adopt to mercury free gold processing methods as highlighted during one of the Focus Group Discussions as shown below.

*"Currently, it's challenging to attract customers to mercury-free (MF) gold mining operations when there are still individuals using mercury to extract gold quickly. The market for mercury-free gold is relatively small compared to the demand for mercury-based gold extraction. As a result, mercury-free sites struggle to attract customers, requiring a larger volume of processed material to turn a profit. This creates a significant barrier for MF operations in terms of market demand." Female participant FGD Busia*

In one of the focus group discussions with the youth as shown in Fig 4, it was highlighted that they consistently use mercury because it can easily help them get the small contents of mercury and also maximizes their economic benefit as indicated below.

*"For new methods, we may not all benefit. E.g., if a miner has 1 basin of ore, he is assured of the UGX 10,000 (~2.75 USD) of panning, so we have to continue using mercury. Even gold is scarce now, you can't get it without mercury" Miner, Youth FGD.*

## Discussion

Generally, the study revealed low adoption of borax and limited training coverage on the use of borax. 80.1% of respondents saved money attained from artisanal small scale gold mining activities, and only 21.5% had investments. Qualitative data revealed that miners often resort to mercury use for quick income as using mercury takes little time compared to mercury-free methods and also the dire need for government intervention and need for policies to promote safer gold processing methods for sustainable livelihoods.

This study purposed to assess the effects of mercury-free gold processing on the incomes and livelihoods of ASGMs and stakeholders in Busia, Buhweju, and Kassanda districts in Uganda.

Artisanal Small-Scale Gold Miners are usually identified with poverty from a sustainability and economic dimension. This economic dimension is closely related to the environmental and social dimensions which justify the increased influx of people who are getting involved in the gold mining sector. We believe that efficient technological interventions and

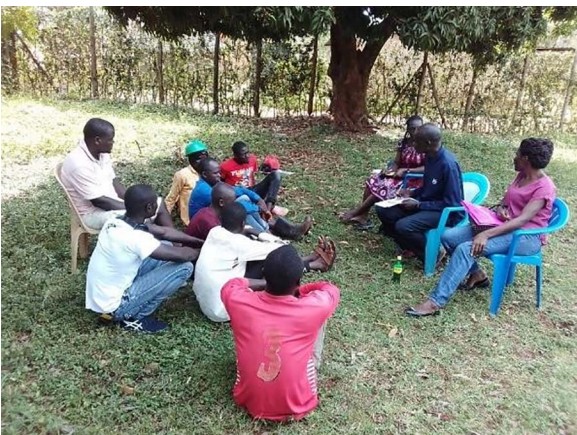

**Fig 4. Youths in a focus group discussion.**

replacing the mercury amalgamation process with an efficient gold-capturing method will have a significant impact on the environment and economic goals.

Many methods of mercury-free mining are available around the globe. The method of using borax has become one of the favorable methods for extracting gold due to its lower operating cost, and higher recovery [25]. Furthermore, the use of borax causes insignificant effects on human health and the environment compared to the use of mercury. The process is shown to possess lower operating costs and higher recovery of gold as a result of no formation of mercury flour which leads to gold losses in the process [25]. Therefore, there are many positive reasons for using borax to extract gold. The mercury-free gold extraction method with borax uses wet milling, yet several mining sites are located on hilly and dry areas usually found far away from sources of water. ASGMs access water by paying exorbitant prices for one jerry can of water [26] which is another barrier to using wet milling.

Artisanal small-scale gold mining is frequently practiced in rural areas by semi-skilled artisans with various social, economic, and technical constraints that impede optimal extraction of mineral deposits. This disadvantaged standpoint position deprives artisanal miners of accessing various social economic opportunities to maximize profits from the mining activity, unlike large-scale mining companies. In short, their disadvantaged position makes them lack the necessary technical–economic information for long-term planning. As a result, frequently artisanal miners experience limited financial resources to invest in essential capital goods and services for efficient mining fueled by often no access to bank loan facilities largely due to lack of loan securities [27,28].

The majority of the respondents in this study were male (77.6%) and the mean age was 32 years, SD + 10.8, minimum age 18, and maximum 73 years. Most miners were predominately male (77.6%). The mean age of miners in this study concurs with findings from Mali and Burkina Faso in 2020 that showed that the majority of Gold miners (50%) are between 25 and 35 years old [29], in Cameroon most miners (38.5%) were the ages 26–35 years [30]. Also, most of them are usually men similar to this study that showed 77.6% of miners were men.

Just like in Cameroon where most miners 33.0% attained the primary level as their highest level of education, miners in this study had low levels of education (36.6% went up to the primary level [30]. This is typical of artisanal miners since artisanal work such as hand-dug pits tends to attract low-skilled, impoverished, low educated, and unemployed people (IOM, 2020). The majority of miners in this study (83%) had worked in the mines for a period between a few months to 10 years, similar to findings from Cameroon with the majority of miners (58.6%) having worked for 0–10 years. Similarly, to other studies, the cost of mercury is reported to be generally cheap. However, this study showed a very low cost of mercury ($0.27 per gram/1000Ugx) at the mining sites compared to that reported in Zimbabwe of $2 per gram [31].

However, borax is still way cheaper than mercury, for instance, 1 g of borax costs about 100/= ($0.027) (though sold in packs of 200g). This study showed a low proportion of miners who use borax (16.1%) but with more efforts on sensitization, there is potential for more miners to adopt borax use. Studies have demonstrated the adoption of the gravity borax method in gold extraction by more than 90% of the miners in some areas [26]. This study however expressed the need to raise more awareness about mercury-free gold mining methods for instance, as less than half of the miners (42.9%) were aware of Borax method and only 21.1% had been trained in the borax method. In this study, even though neither the mercury nor Borax use method was associated with the amount of money earned by the miners, the majority of the miners (80%) save money earned from mining activities which also they use to make investments such as buy land, business projects, support children 's education and personal developments.

However, the average monthly income from mining activities reported in this study (UGX 436,770/ $114.06) is higher than that reported from Cameroon ($22.4/15,000 FCFA) [32]. This implies that the opportunity to promote mercury free gold mining methods can still be explored among the ASGM to further boost their incomes. Mercury is used in more than 70 countries globally to extract gold in artisanal small-scale gold mining (ASGM). The application of mercury is simple and plays a key role in the livelihood for more than 16 million gold miners. Various studies conducted on artisanal small-scale gold mining have revealed that most significantly the widespread use of mercury for gold extraction results in polluted

environments and serious health hazards for the miners themselves and for the population in the vicinity of small-scale gold mining settlements. This results into loss of lives and increases treatment costs for the miners which reduces their income and livelihoods. Large amounts of mercury are transferred to the environment from small-scale gold mining activities in Uganda [33].

Several alternative methods have been suggested and tested with limited degrees of success [34]. In 2009 Geocenter Denmark financed a project to test the feasibility of using borax as a replacement for mercury in small-scale gold extraction in Tanzania. We believe that the efficient technological interventions and replacing the mercury amalgamation process with an efficient gold capture method will have a significant impact on the environment and economic goals.

In the Free Your Mine 2019–2021 project ASGMs were introduced to the economic and health advantages of the mercury-free gold extraction method as an incentive to change to the mercury-free method. The strengths of the mercury-free gold extraction method include low costs, higher gold yield, benign environmental impact, and legality. However, the mercury-free method may be slightly more time-consuming than the amalgamation method, especially in the beginning [10].

The use of chemical borax, also known as sodium borate, appears to be one of the more viable ways that have been proposed to reduce or stop the use of mercury by small-scale miners [35]. Borax is used for cleaning purposes and is therefore commonly available. The reason for using borax in the smelting process of ore material is that borax reduces the melting point of metals and minerals. Under normal field circumstances, small-scale miners cannot smelt gold, as they cannot create the high temperature required to smelt the ore. By adding borax to their concentrate, however, they can extract and smelt their gold. Gold purchasers already use borax to purify gold with a high content of mercury; however, the method has only recently been applied by small-scale miners. In the Benguet area of the northern Philippines, around 15,000 small-scale gold miners currently mix their gold concentrate with borax, followed by heating and smelting [36,37].

Changing a well-established habit is difficult and requires very good reasons. The habit of using mercury for gold extraction is such a case because the mercury method is easy to learn and carry out. This makes the introduction of a new and healthier method a challenge, especially because the borax method requires skill and patience. Depending on the skills of the person who prepares the concentrates, the borax method may take between half an hour and an hour longer than the mercury method. The immediate advantage of using borax is that it does not harm the environment or the people within or close to the mining sites, in the quantities necessary for gold extraction. In addition, borax is cheaper than mercury and produces purer gold than that produced with mercury. As small-scale miners are often paid according to the gold content in their gold, the gold produced with borax is likely to provide a better price. However, the question remains whether these advantages are sufficient to make small-scale gold miners swap from mercury to borax. Considering the embedded culture of using mercury, a change from mercury to borax extraction is not likely to come easily. An additional incentive for the miners to convert to the borax method may be needed.

## Conclusions

The study underscores the urgent need to transition from mercury-based gold extraction to mercury-free methods in artisanal and small-scale gold mining (ASGM) in Uganda given its positive impact on the environment, income and livelihood of miners. The findings demonstrate that the use of the borax method not only offers economic benefits, including lower operating costs and higher gold yields, but also has significant environmental and health advantages. Moreover, the study emphasizes the socio-economic context of ASGMs, including their limited access to education and financial resources, which can hinder their ability to adopt new technologies and practices. Therefore, interventions aimed at promoting borax use should also include capacity-building initiatives and financial support for ASGMs.

On the one hand, most miners recognized the potential of using the borax method, as it would significantly reduce mercury usage and improve the purity of their gold. On the other hand, they mentioned the high price of acetylene gas, the fact that borax is not readily available in the region (this is true and ways of making borax available in that part of the country are needed), the longer time required for preparing the concentrate, and the fact that many miners need to

process small quantities of gold (e.g., 0.3 gram) daily to get food on the table and that the borax method seemed a bit too advanced for such small quantities.

The borax method was received with interest in both small-scale gold-mining communities, but the attitude was more positive in the more permanent settlement of Buhweju where people are more concerned about the environment. The following points must be taken into account to ensure increased uptake of borax; (1) Locally produced, inexpensive blowers or acetylene gas must be easily available, as well as access to the necessary expertise, (2) borax must be readily available, (3) a substantial training program has to be carried out, (4) a link must be established between the small-scale miners and advisers, preferably the local mining authorities, who can guide the miners when technical problems occur and (5) small-scale miners need to understand the link between the borax method and a higher gold recovery rate.

The study also highlights the challenges in adopting mercury-free methods, such as the need for access to borax and water for wet milling, as well as the lack of awareness and training among miners. Addressing these challenges will require collaborative efforts from government agencies, NGOs, and other stakeholders to provide support and resources to ASGMs.

Generally, the findings of this study emphasize the importance of promoting and supporting the transition to mercury-free gold extraction methods in ASGMs in Uganda. By doing so, we can not only improve the incomes and livelihoods of ASGMs but also protect the environment and health of miners and surrounding communities.

## Supporting information

**S1 Appendix. Minimal dataset.**
(XLSX)

## Acknowledgments

The completion of this study was through the invaluable efforts of many stakeholders. The study team is grateful to the Project Officers in all three districts, the study team, and research assistants for conducting the study. Our thanks also go to the district leaders, all the district local government officials, mining organizations and the artisanal small-scale gold miners and Community members for their shared information.

## Author contributions

**Conceptualization:** James Natweta Baguma, Victoria Mukasa, D.K Sekimpi, Daniel Sekabojja, Kamese Geoffrey, Eva Magambo, John Ssempebwa, Margrethe Smidth.

**Data curation:** James Natweta Baguma, Victoria Mukasa, D.K Sekimpi, Daniel Sekabojja, Victoria Nabankema, John Ssempebwa, Margrethe Smidth.

**Formal analysis:** James Natweta Baguma, D.K Sekimpi, Daniel Sekabojja, Victoria Nabankema, Kamese Geoffrey, Eva Magambo, John Ssempebwa, Margrethe Smidth.

**Funding acquisition:** James Natweta Baguma, D.K Sekimpi, Daniel Sekabojja, Victoria Nabankema, Margrethe Smidth.

**Investigation:** James Natweta Baguma, D.K Sekimpi, Daniel Sekabojja, Victoria Nabankema, Kamese Geoffrey, Margrethe Smidth.

**Methodology:** James Natweta Baguma, D.K Sekimpi, Victoria Nabankema.

**Project administration:** James Natweta Baguma, Kamese Geoffrey, Eva Magambo.

**Resources:** James Natweta Baguma, Victoria Mukasa, Eva Magambo.

**Software:** James Natweta Baguma, Eva Magambo.

**Supervision:** James Natweta Baguma, Kamese Geoffrey, Eva Magambo.

**Validation:** James Natweta Baguma, Eva Magambo.

**Visualization:** James Natweta Baguma.

**Writing – original draft:** James Natweta Baguma.

**Writing – review & editing:** James Natweta Baguma, John Ssempebwa.

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
