## [Decision Letter · Decision Letter 0]

30 Mar 2025

Dear Dr. Baguma,

Thank you for submitting your manuscript to PLOS ONE. After careful consideration, we feel that it has merit but does not fully meet PLOS ONE’s publication criteria as it currently stands. Therefore, we invite you to submit a revised version of the manuscript that addresses the points raised during the review process.

We look forward to receiving your revised manuscript.

Kind regards,

Vinaya Satyawan Tari, Post doctoral fellow, (M.Sc., B.Ed., Ph.D.)

Academic Editor

PLOS ONE

Journal Requirements:

2**.** We note that the grant information you provided in the ‘Funding Information’ and ‘Financial Disclosure’ sections do not match.

“The Uganda National Association of Community and Occupational Health (UNACOH), with 710 funding support from Dialogos, a Danish Non-Government Organization supported by Danish 711 Government conducted a study in Buhweju, Kassanda and Busia districts under the Free Your 712 Mine project (FYM 2025) to assess effects of mercury-free gold processing on the incomes 713 and livelihoods of artisanal and small-scale gold miners and stakeholders in Buhweju, Busia 714 and Kassanda districts in Uganda.”

“The author(s) received no specific funding for this work”

5. Please ensure that you refer to Figure 1-3 in your text as, if accepted, production will need this reference to link the reader to the figure.

6. We note that Figure 1-3 includes an image of a participant.

Reviewers' comments:

Reviewer's Responses to Questions

**Comments to the Author**

1. Is the manuscript technically sound, and do the data support the conclusions?

Reviewer #1: Yes

Reviewer #2: Partly

Reviewer #3: Partly

2. Has the statistical analysis been performed appropriately and rigorously?

Reviewer #1: Yes

Reviewer #2: No

Reviewer #3: I Don't Know

3. Have the authors made all data underlying the findings in their manuscript fully available?

Reviewer #1: Yes

Reviewer #2: Yes

Reviewer #3: Yes

4. Is the manuscript presented in an intelligible fashion and written in standard English?

Reviewer #1: Yes

Reviewer #2: Yes

Reviewer #3: No

Reviewer #1: Article is well structured, well written and technically sound. Authors should simply check again for any minor grammatical errors and correct them. e.g. line 337 and 338 (change 'its' to it; 'so' into to).

Reviewer #2: The manuscript comes out as an advocacy for the use Borax in gold processing, without much justification. Borax itself has not been described/ defined and there does not appear to be any significant technical references on the efficacy of Borax. There is clearly limited uptake of Borax with artisanal miners in the study preferring the use of mercury in gold processing, instead. One quoted statistic says that in a field demonstration, Borax recovered 40% more gold than mercury, but the experiment itself is not properly described. More examples of properly set-up experimentations, or of actual mine recovery statistics may help lend more objectivity to the study. It appears the stakeholders are saying that mercury is a faster and cheaper way of gold recovery than mercury. However, even in the discussion, the authors hold on to a 'truth' that Borax use is 'less time-consuming'.

The tone of the manuscript throughout should refrain from obvious biases and be more objective; the authors should let the data speak for themselves.

Although generally well-written, the manuscript has some typographic errors (e.g. P in sample size calculation is applied as 12%, but stated as 50% in the list of variables) and repetitiveness (e.g. in the Discussion) which need to be rectified throughout.

Reviewer #3: General Comment:

This study represents a substantial mixed-methods effort addressing a globally significant issue: the use of mercury by artisanal gold miners—a metal highly toxic to both human health and the environment.

However, the manuscript requires a thorough revision of the presentation of results and a structural reworking of the article to make it clearer and more accessible to readers.

The stated objective of the article is to “assess the impact of borax utilization in gold processing on the income and livelihoods of miners and stakeholders in Buhweju, Busia and Kassanda districts in Uganda where the Free Your Mine project is being implemented.”

However, the content of the article suggests that it primarily evaluates the barriers and facilitators to borax use within this population. Therefore, we recommend that the objective be reformulated to more accurately reflect the actual focus of the study.

Additionally, terminology should be used consistently throughout the manuscript:

• borax should be written in lowercase;

• Abbreviations should not be defined multiple times.

Introduction:

• Clearly present, based on the literature, the benefits and drawbacks of using borax.

• The "Free Your Mine" project needs to be better explained: objectives, context, and implementation sites.

• The data from Ghana, mentioned later in the article, would be more appropriately integrated into the discussion section, where they can help contextualize the findings.

Methods:

• Add a map to indicate the inclusion areas.

• Specify the number of participants per site.

• Indicate how many participants were trained in mercury-free techniques.

• Review the mathematical formulas:

o The “P” used does not refer to the prevalence as claimed.

o It would be preferable to clearly explain the calculation in plain language.

• Provide more context on ASGM (artisanal and small-scale gold mining) in Uganda: are these activities legal or not?

• Lines 173–176: it is unclear whether the KIIs involved 12 groups or 5.

• Define what is meant by "youth mining groups."

• The sampling method needs to be explained more clearly.

• Line 205: specify whether the qualitative questionnaires were structured or semi-structured, and include the questionnaire in an appendix.

Results:

• Lines 266–269 should be removed.

• The denominators used for ratio calculations must be clearly presented. Since some questions were conditional, readers need to follow how the calculations were made.

• Table 4 should show the sample size in each category.

• All table titles should be placed above the tables.

• Clearly separate descriptive data from the quantitative and qualitative components. For instance, the profile of focus group participants is unclear.

• Line 296: 21.7% does not constitute a “notable majority.”

• Line 299: 1.9% of participants use borax for gold extraction — which equates to only 3 individuals? Clarify the distinction between purification and extraction, since the key issue is replacing mercury in the extraction process.

o How were participants categorized as borax users or not with such a low number? This represents a major bias in the study.

o Hence the need to reformulate the study objective accordingly.

• A flow chart should be added, showing:

o Number of participants trained vs. untrained;

o Their behavior in terms of borax use vs. mercury or other methods.

• Assess whether borax use varies depending on the location (e.g., proximity to the Free Your Mine project site).

• Data on borax or mercury use should be accompanied by a time frame/unit (cf. Line 301, Table 3, Line 390, Table 8...).

• Present monetary amounts in Ugandan shillings with USD equivalents.

• Line 322: the interpretation provided does not match the content of the referenced table, which is titled “Likelihood of continuing to use mercury after learning about gravity/borax method.”

• Barriers to borax use are not presented in the results section, despite being a central takeaway of the study. They are only discussed later, which is a limitation.

• Line 364: the table does not match the title.

• Line 366: check for possible errors in the standard deviation (SD).

• Line 388: the phrase “accepted that” is unclear — please rephrase.

• Line 392: clarify what other methods (besides mercury or borax) are being referred to.

• Table 8: if a respondent used mercury more than 12 years ago, should they still be counted as a current user?

Discussion:

• Start with a brief summary of the main findings, before discussing them.

• Clearly present the limitations of the study.

• Discuss the results in relation to the regional context and existing literature.

• The literature-based benefit-risk analysis of borax should be placed in the introduction, not in the discussion.

• Compare your findings with those from other countries, such as Ghana.

• Line 435: not clearly worded — rephrase.

• Lines 441–442: these elements should appear in the introduction.

• Lines 462–464: same comment.

• Lines 475–495: limited relevance — this section could be condensed to two sentences.

• Lines 558–562: it is uncertain whether these conclusions are fully supported by the results.

• Line 572: clarify what is meant by "acetylen gas", or remove the reference.

Finally, the article should clearly discuss the barriers to borax adoption, as well as measures to engage communities in transitioning to mercury-free gold mining.

Is the main issue access to borax, water availability, lack of training, or time required for processing? These points should be clarified.

Conclusion:

The article would benefit greatly from a clear reorganization, removal of redundancies, and better alignment between the stated objectives and the results actually presented.

**Do you want your identity to be public for this peer review?** For information about this choice, including consent withdrawal, please see our Privacy Policy

Reviewer #1: No

Reviewer #2: **Yes: ** Antony Mamuse

Reviewer #3: No

---

## [Author Response · Author response to Decision Letter 1]

4 Apr 2025

Dear Reviewers,

Thank you so much for your speedy consideration of our manuscript. We have to the best of our ability tried to rectify according to your valued comments from the reviewers.

I am enclosing herewith a revised manuscript entitled “Utilization of borax and its impact on the income and the livelihood of miners and other stakeholders: A case of Uganda.”

Please, find below our comments summarized in a table attached.

I look forward to your positive response.

Regards

---

## [Editor Report · Decision Letter 1]

17 Apr 2025

Dear Dr. Baguma,

We look forward to receiving your revised manuscript.

Kind regards,

Vinaya Satyawan Tari, Post doctoral fellow, (M.Sc., B.Ed., Ph.D.)

Academic Editor

PLOS ONE

---

## [Author Response · Author response to Decision Letter 2]

30 Apr 2025

RE: SUBMISSION OF THE REVISED MANUSCRIPT TO YOUR JOURNAL

Thank you so much for your speedy consideration of our manuscript. We have to the best of our ability tried to rectify according to your valued comments from the reviewers.

I am enclosing herewith a revised manuscript entitled “Utilization of borax and its impact on the income and the livelihood of miners and other stakeholders : A case of Uganda.”

This study was funded by Dialogos, a non-governmental organization (NGO) based in Denmark. The support from Dialogos was instrumental in enabling the successful implementation of the project activities.

Please, find below our comments summarized in a table.

I look forward to your positive response.

Sincerely,

Mr. James Natweta Baguma

Research Associate

Department Disease Control and Environmental Health.

Makerere University School of Public Health

Tel: +256775989895 Email: bagumajamesnat@gmail.com

---

## [Decision Letter · Decision Letter 2]

10 Jun 2025

Dear Dr. Baguma,

Thank you for submitting your manuscript to PLOS ONE. After careful consideration, we feel that it has merit but does not fully meet PLOS ONE’s publication criteria as it currently stands. Therefore, we invite you to submit a revised version of the manuscript that addresses the points raised during the review process.

We look forward to receiving your revised manuscript.

Kind regards,

Vinaya Satyawan Tari, Post doctoral fellow, (M.Sc., B.Ed., Ph.D.)

Academic Editor

PLOS ONE

Reviewers' comments:

Reviewer's Responses to Questions

**Comments to the Author**

Reviewer #2: (No Response)

2. Is the manuscript technically sound, and do the data support the conclusions?

Reviewer #2: Partly

3. Has the statistical analysis been performed appropriately and rigorously?

Reviewer #2: Yes

4. Have the authors made all data underlying the findings in their manuscript fully available?

Reviewer #2: Yes

5. Is the manuscript presented in an intelligible fashion and written in standard English?

Reviewer #2: Yes

Reviewer #2: Although in the summary table of responses, the authors indicate that they have adequately responded to concerns raised earlier, I have not been able to verify that the authors have actually:

1. Defined borax and explained its use and adequately corroborated its efficacy

2. Provided details of a specific experiment which supposedly showed that borax is 40% more efficient than mercury in gold recovery.

3. Adequately replaced advocacy with objectivity throughout the article.

**Do you want your identity to be public for this peer review?** For information about this choice, including consent withdrawal, please see our Privacy Policy

Reviewer #2: **Yes: ** Antony Mamuse

---

## [Author Response · Author response to Decision Letter 3]

12 Jun 2025

Editor

PLOS ONE

RE: SUBMISSION OF THE REVISED MANUSCRIPT TO YOUR JOURNAL

Thank you so much for your speedy consideration of our revised manuscript. We have to the best of our ability tried to rectify according to your valued comments from the reviewers.

I am enclosing herewith a revised manuscript entitled “Utilization of borax and its impact on the income and the livelihood of miners and other stakeholders: A case of Uganda.”

This study was funded by Dialogos, a non-governmental organization (NGO) based in Denmark. through the Uganda National Association of Community and Occupational Health. (UNACOH). The support from Dialogos was instrumental in enabling the successful implementation of the project activities.

Please, find below response summarized in a table.

I look forward to your positive response.

Sincerely,

Mr. James Natweta Baguma

Research Associate

Department Disease Control and Environmental Health.

Makerere University School of Public Health

Tel: +256775989895 Email: bagumajamesnat@gmail.com

REMARKS STATUS LOCATION

1. Defined borax and explained its use and adequately corroborated its efficacy We have defined and explained borax and its use as well as its efficacy Line 76-79

Line 81-89

2. Provided details of a specific experiment which supposedly showed that borax is 40% more efficient than mercury in gold recovery. The details of this experiment are well highlighted in the study conducted by Stoffersen B et al titled Comparison of gold yield with traditional amalgamation and direct smelting in artisanal small-scale gold mining in Uganda. Journal of Health and Pollution. 2019 Dec 1;9(24):191205. Reference 8

3. Adequately replaced advocacy with objectivity throughout the article. We have adjusted the language Throughout the manuscript

---

## [Decision Letter · Decision Letter 3]

18 Jun 2025

Utilization of borax and its impact on the income and the livelihood of miners and other stakeholders: A case of Uganda.

PONE-D-25-08547R3

Dear Dr. Baguma,

We’re pleased to inform you that your manuscript has been judged scientifically suitable for publication and will be formally accepted for publication once it meets all outstanding technical requirements.

Kind regards,

Vinaya Satyawan Tari, Post doctoral fellow, (M.Sc., B.Ed., Ph.D.)

Academic Editor

PLOS ONE

Reviewers' comments:

Reviewer's Responses to Questions

**Comments to the Author**

Reviewer #2: All comments have been addressed

2. Is the manuscript technically sound, and do the data support the conclusions?

Reviewer #2: Yes

3. Has the statistical analysis been performed appropriately and rigorously?

Reviewer #2: Yes

4. Have the authors made all data underlying the findings in their manuscript fully available?

Reviewer #2: Yes

5. Is the manuscript presented in an intelligible fashion and written in standard English?

Reviewer #2: Yes

Reviewer #2: The authors have explained what borax is and pointed to a study which specifically assessed the efficiency of borax vs mercury in gold ore processing. The borax advocacy tone has been largely rectified.

**Do you want your identity to be public for this peer review?** For information about this choice, including consent withdrawal, please see our Privacy Policy

Reviewer #2: **Yes: ** Antony Mamuse

---

## [Editor Report · Acceptance letter]

PONE-D-25-08547R3

PLOS ONE

Dear Dr. Baguma,

I'm pleased to inform you that your manuscript has been deemed suitable for publication in PLOS ONE. Congratulations! Your manuscript is now being handed over to our production team.

Kind regards,

on behalf of

Dr. Vinaya Satyawan Tari

Academic Editor

PLOS ONE